# Minimally Invasive Postero-Inferior Sacroiliac Joint Fusion: Surgical Technique and Procedural Details

**DOI:** 10.3390/jpm13071136

**Published:** 2023-07-14

**Authors:** Usman Latif, Paul J. Hubbell, Goran Tubic, Luis A. Guerrero, Ioannis M. Skaribas, Jon E. Block

**Affiliations:** 1Department of Anesthesiology, Pain and Perioperative Medicine, The University of Kansas Hospital, 4000 Cambridge St., Kansas City, KS 66160, USA; 2Southern Pain and Neurologic, 3348 W Esplanade Ave., Ste. A, Metairie, LA 70002, USA; 3Chicagoland Pain Management, 420 S. Schmidt Rd., Ste. 110, Bolingbrook, IL 60440, USA; 4Central Florida Pain Relief Centers, 683 Douglas Ave., Ste. 101, Altamonte Springs, FL 32714, USA; 5Expert Pain, 11451 Katy Fwy., Ste. 340, Houston, TX 77079, USA; 6Independent Consultant, 2210 Jackson St., Ste. 401, San Francisco, CA 94115, USA

**Keywords:** sacroiliac, fusion, intra-articular, minimally invasive

## Abstract

(1) Background: Minimally invasive sacroiliac joint (SIJ) fusion is the preferred surgical intervention to treat chronically severe pain associated with SIJ degeneration and dysfunction. (2) Methods: This paper details the ten-step surgical procedure associated with the postero-inferior approach using the PsiF™ DNA Sacroiliac Joint Fusion System. (3) Results: The posterior surgical approach with an inferior operative trajectory (postero-inferior) utilizes easily identifiable landmarks to provide the safest, most direct access to the articular joint space for transfixing device placement. Implanting the device through the subchondral bone provides maximum fixation and stabilization of the joint by utilizing an optimal amount of cortical bone–implant interface. Approaching the joint from the inferior trajectory also places the implant perpendicular to the S1 endplate at a “pivot point” near the sacral axis of rotation, which addresses the most significant motion of the joint. (4) Conclusions: Further observational data from real-world clinical use are encouraged to further validate this procedure as the surgical preference for minimally invasive SIJ fusion.

## 1. Introduction

The sacroiliac joint (SIJ) is recognized as a potent pain generator that can act as both a primary and a contributing source of chronically severe low back pain [1,2,3]. Sacroiliitis can also manifest as adjacent segment disease secondary to lumbar spinal fusion surgery [4,5,6]. When symptoms related to SIJ dysfunction become unresponsive to conservative care and impair normal physical function and quality of life, minimally invasive SIJ arthrodesis is the preferred surgical option [7]. Surgical fixation and stabilization of the SIJ to support bony fusion across the joint space can be accomplished using an array of surgical approaches [8,9,10]. While the surgical goal of all approaches is identical, the posterior approach utilizes a trajectory and easily identifiable landmarks that allow the surgeon to control the risk of violating important neuro-vascular structures [11].

The minimally invasive, transfixing SIJ fusion procedure can employ numerous operative trajectories. To date, the most common trajectories utilized have been lateral and lateral-oblique trajectories. However, these operative trajectories come with certain risks and pitfalls including, most commonly, implant breech of the neuroforamen or anterior sacral cortex [12,13,14]. Furthermore, the implant location most associated with complications is placed in the superior aspect of the joint [12]. In contrast, the postero-inferior trajectory allows implant placement in an ultra-safe operative corridor, originating at the inferior aspect of the joint, going through the ilium, across the sacroiliac joint space, and into the sacrum, transfixing the osseous confines of the ilium and sacrum across the sub-chondral bone.

## 2. Materials and Methods

This paper details the surgical procedure associated with the postero-inferior operative trajectory (*Omnia* postero-inferior approach) using the PsiF™ DNA Sacroiliac Joint Fusion System [15]. Included in this description will be a discussion of the biomechanical advantages of placing the Dorsal Nutation Anchor (DNA) in close proximity to the predominant rotational axis and utilizing natural joint architecture to stabilize the fusion construct.

## 3. Results

### 3.1. Structural Anatomy Overview

Figure 1 illustrates the important anatomical landmarks and features of the SIJ in perspective to the overall pelvic anatomy. Auricular-shaped surfaces encompass both the ilio-sacral joint aspects. As the primary structural connection between the axial and lower appendicular skeleton, the SIJ includes load transfer, weight-bearing, and shock absorption characteristics, but has limited range of motion [16].

### 3.2. Patient Selection

The PsiF™ Sacroiliac Joint Fusion System is intended for SIJ fusion for conditions including degenerative sacroiliitis and sacroiliac joint disruptions. Minimally invasive fusion of the SIJ is considered medically necessary by most insurance plans when ALL of the following criteria are met:Moderate to severe pain with functional impairment and pain persists despite a minimum of 6 months of intensive nonoperative treatment that must include medication optimization, activity modification, bracing, and active therapeutic exercise targeted at the lumbar spine, pelvis, SIJ, and hip including a home exercise program.A patient that reports typically unilateral pain that is caudal to the lumbar spine (L5 vertebrae), localized over the posterior SIJ, and consistent with SIJ pain.A thorough physical examination demonstrating localized tenderness with palpation over the sacral sulcus (Fortin’s point) in the absence of tenderness of similar severity elsewhere (e.g., greater trochanter, lumbar spine, and coccyx) and that other obvious sources for their pain have been ruled out.Positive response to a cluster of 3 provocative tests (e.g., thigh thrust test, compression test, Gaenslen’s test, distraction test, FABER test, and posterior provocation test).Absence of generalized pain behavior or generalized pain disorders (e.g., fibromyalgia) contributing to the SIJ-area pain.Diagnostic imaging studies that include ALL of the following:−Imaging (plain radiographs and a CT or MRI) of the SIJ that excludes the presence of destructive lesions (e.g., tumor, infection), fracture, traumatic SIJ instability, or inflammatory arthropathy that would not be properly addressed by percutaneous SIJ fusion;−Imaging of the pelvis (AP plain radiograph) to rule out concomitant hip pathology;−Imaging of the lumbar spine (CT or MRI) to rule out neural compression or other degenerative condition that can be causing low back or buttock pain.


Additionally, the patient should experience:At least a 75 percent reduction of pain for the expected duration of a standard anesthetic agent, and the ability to perform previously painful maneuvers, following an image-guided, contrast-enhanced intra-articular SIJ injection;A trial of at least two or more intra-articular SIJ injections, with at least one injection being therapeutic (i.e., corticosteroid injection).

### 3.3. Surgical Technique

Pre-operative planning for this procedure consists of lateral and oblique pelvic radiographs, as well as fluoroscopic inlet views (20–25° caudally) and AP/outlet views (35° outlet) to identify important SIJ anatomical landmarks.

For optimal intra-operative imaging, the fluoroscope should be equipped to rotate around the operating table 30–35° vertically. Postero-inferior SIJ trans-fixation fusion surgery is performed minimally invasively under fluoroscopic guidance, and the following provides the operative details on a step-by-step basis:

#### 3.3.1. Patient Positioning

Using a radiolucent table (flattop or Jackson table), the patient is placed in the prone position with the lumbar spine optimally flexed to minimize lumbar lordosis. Then, 6″–8″ gel packs or a blanket roll under the umbilicus can be used to elevate the lumbar spine out of lordosis. This allows the hips to have approximately 15–20° of flexion. Positioning the patient to achieve a flat back helps to remove pelvic tilt and provides for key SIJ landmark identification under fluoroscopy. The procedure should be performed in a standard operative environment in a sterile surgical field.

#### 3.3.2. Intraoperative Imaging

Once the patient is properly positioned, fluoroscopic imaging should be undertaken to obtain lateral, oblique, inlet, and AP/outlet views to identify the SIJ and its anatomical extent (Figure 2).

#### 3.3.3. Approach and Incision

The operative goal of postero-inferior SIJ fusion is to place the Dorsal Nutation Anchor Implant in a minimally invasive fashion, originating inferior to the posterior superior iliac spine on a trajectory that allows the implant to transfix the osseous confines of the ilium and sacrum, as closely adjacent to the sacral x-axis as possible to allow for maximum fixation while minimizing the biomechanical forces on the implant [17].

After the patient has been placed in the proper prone position, the initial surgical step is to draw a vertical line on the skin over the midline portion of the sacrum and position the C-arm of the fluoroscope over the sacrum in direct AP view to identify the key landmarks such as the SIJ and sacral cornua, as well as the S1, S2, and S3 foramen. Next, a K-wire should be placed horizontally to the SIJ, directly over each sacral horn, and the skin should be marked on the vertical midline representing the sacral cornua. Skin marking should then be repeated for the S1, S2, and S3 foramen.

For unilateral procedures, the incision will start lateral to the apex of the posterior superior iliac spine, and the incision should also be slightly lateral to the postero-inferior joint line. The incision should extend inferiorly to the inferior aspect of the SI joint. A knife and/or bovie should be used to dissect tissue down to the ilium and finger palpation may be used for blunt tissue dissection to the inferior aspect of the posterior superior iliac spine in order to confirm a clear path to the implantation site.

#### 3.3.4. Steinman Pin Placement

Under fluoroscopic guidance using the inlet view, advance a Steinmann pin starting inferior, ventral, and lateral to the PSIS, and advancing superiorly, through the ilium across the SI joint into the sacrum. Finally, use a lateral fluoroscopic view to verify that the tip of the Steinmann pin has passed the posterior cortical outline of the sacrum, and is docked in the sacrum at or near the center of the second sacral body (Figure 3 and Figure 4).

#### 3.3.5. Tissue Dilation

With AP/outlet view visualization, a scalpel should be used to create a 2–3 cm skin vertical incision lateral to the postero-inferior joint line, at the apex of the posterior superior iliac spine, extending inferiorly to the inferior aspect of the joint line. The initial dilator instrument should be passed over the K-wire and, using a mallet, advanced until it docks inferior to the posterior superior iliac spine. Use an inlet view to confirm that the angle of the initial dilator is optimal for the transfixion of the SIJ. To confirm proper placement, verify visually via lateral fluoroscopy that both corners of the initial dilator are past the posterior cortical outline and inferior aspect of the ilium (Figure 5). Note: the initial dilator should be secure enough to stand on its own.

#### 3.3.6. Implant Preparation

A mechanically solid implant construct is imperative to the short-term stabilization of the joint. As such, a proper implant bed must be prepared for the construct, utilizing a reamer drill. Oblique and inlet views should be used to confirm the optimal trajectory of instrumentation. There must be adequate room ventral to the instrumentation to ensure that the reamer drill does not violate the anterior border of the inferior joint line (Figure 6). Utilizing inlet and lateral views, an implant bed should be created by advancing the reamer drill over the initial dilator and reaming down to the dense cortical bone of the ilium (Figure 6). The tip of the reamer drill should pass the posterior-inferior border of the ilium in lateral view to ensure that the implant bed is properly prepared.

#### 3.3.7. Docking Washer Introduction

The docking washer should be introduced over the initial dilator and, using inlet and outlet views, should be advanced with a trajectory that allows for transfixion of the joint space by checking the angle of the approach (Figure 7 and Figure 8). To ensure safety, with an inlet view, the ventral aspect of the docking tangs should be observed to have adequate room on the ventral aspect of the joint (Figure 6). Finally, using a lateral view, the impact cap should be attached and advanced using a mallet until the base of the washer tower is seated against the ilium (Figure 6). To complete this step, the impact cap, Initial dilator, and Steinmann pin should be removed, leaving the docking washer tower in place.

#### 3.3.8. Implant Insertion

Once the docking washer has been firmly seated in the inferior aspect of the ilium and proper trajectory has been obtained, the PsiF™ DNA implant is inserted down the docking washer tower. Once the distal tips of the implant make contact with the ilium, ventral pressure is applied as the inserter is turned clockwise. The self-tapping, autograft harvesting implant will advance through the ilium, across the SI joint and into the sacrum (Figure 9 and Figure 10). When final placement is achieved, the base of the implant will lock in the docking washer. The base of the docking washer provides added safety from the foraminal and ventral cortical breech.

#### 3.3.9. Final Implant Placement

Once the PsiF™ DNA implant is optimally placed to provide transfixion and stabilization, the tabs of the docking washer tower can be easily detached from the implant construct. All instrumentation can be removed and final imaging can be conducted (Figure 11). The final implant placement should show the implant securely within the osseous confines of the ilium and sacrum, with the proximal end of the implant and the docking washer docked firmly in the ilium and the distal end of the implant in the sacrum. The incision should be closed in a routine manner. Figure 12 provides a visual guide of all instrumentation used in this minimally invasive SIJ fusion procedure.

## 4. Discussion

The primary surgical objective of all minimally invasive SIJ fusion procedures is to provide immediate fixation and stabilization across the joint space to support osseous consolidation and the development of mechanically solid arthrodesis [18]. This can be accomplished via several different surgical approaches. This paper provides surgical and procedural details regarding the postero-inferior approach which utilizes an operative trajectory that significantly limits the risk of violating the critical neuro-vascular structures by utilizing the ultra-safe corridor of dense cortical bone across the purely articular portion of the SIJ. With reduced peri-operative injury risk, this minimally invasive approach offers a straightforward technique for a physician to learn.

The postero-inferior approach also takes advantage of the natural SIJ architecture by positioning the implant perpendicular to the S1 endplate, inferior and ventral to the PSIS, and near the sacral axis of rotation. This allows for the establishment of a natural “pivot point” around the implant that acts to attenuate the mechanical forces associated with sacral rotation and flexion–extension.

This procedure is accomplished completely under fluoroscopic guidance with minimal tissue disruption. Total procedural time can vary depending on whether the procedure is unilateral or bilateral and the number of implants. Further observational data from real-world clinical use are encouraged to further validate this procedure as the surgical preference for minimally invasive SIJ fusion.

## Figures and Tables

**Figure 1 jpm-13-01136-f001:**
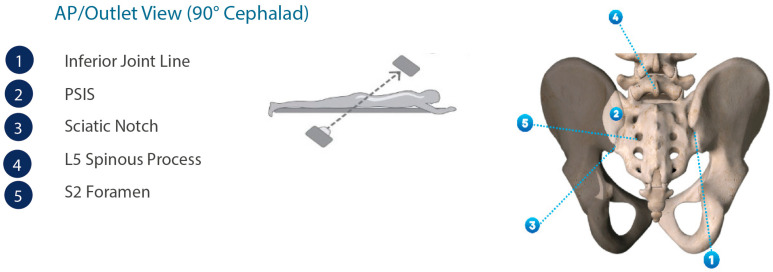
Anatomical renderings of the pelvic anatomy in the AP/outlet view.

**Figure 2 jpm-13-01136-f002:**
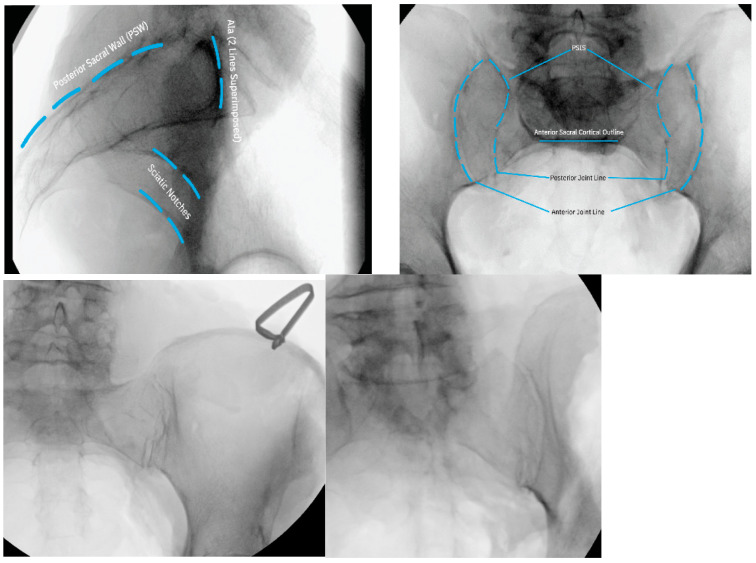
Intraoperative fluoroscopic imaging prior to incision indicating the location and extent of the SIJ.

**Figure 3 jpm-13-01136-f003:**
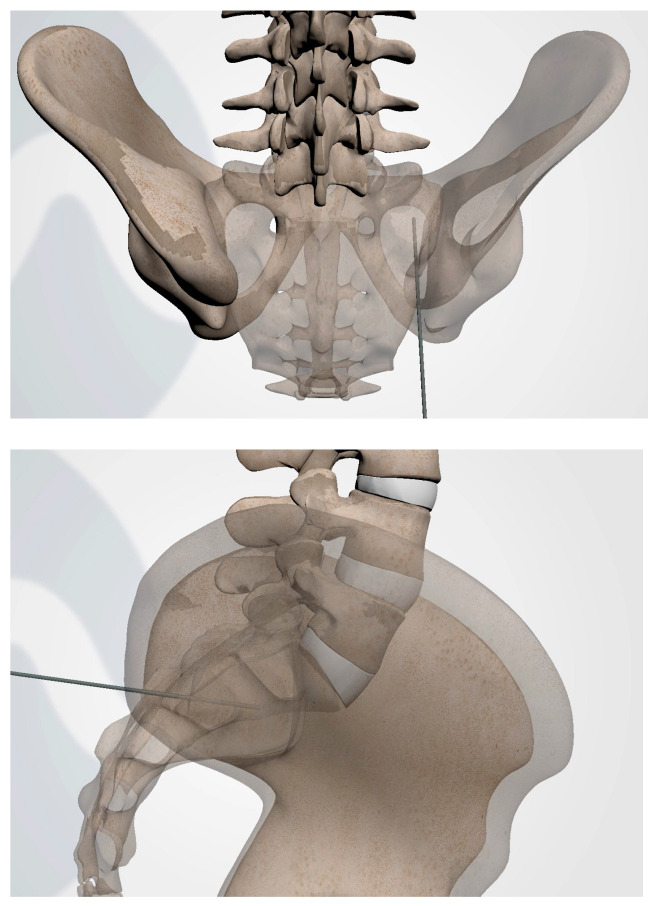
AP (**upper**) and lateral (**lower**) view schematic renderings of proper Steinmann pin introduction and placement.

**Figure 4 jpm-13-01136-f004:**
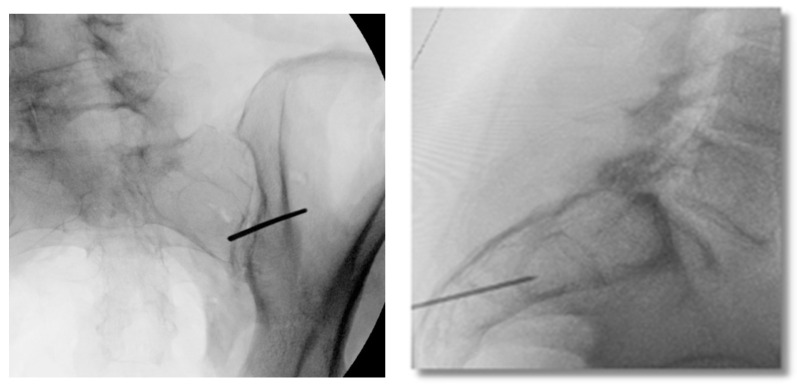
Oblique (**left**), and lateral (**right**) fluoroscopic views indicating the initial proper introduction and placement of the Steinmann pin into the SIJ.

**Figure 5 jpm-13-01136-f005:**
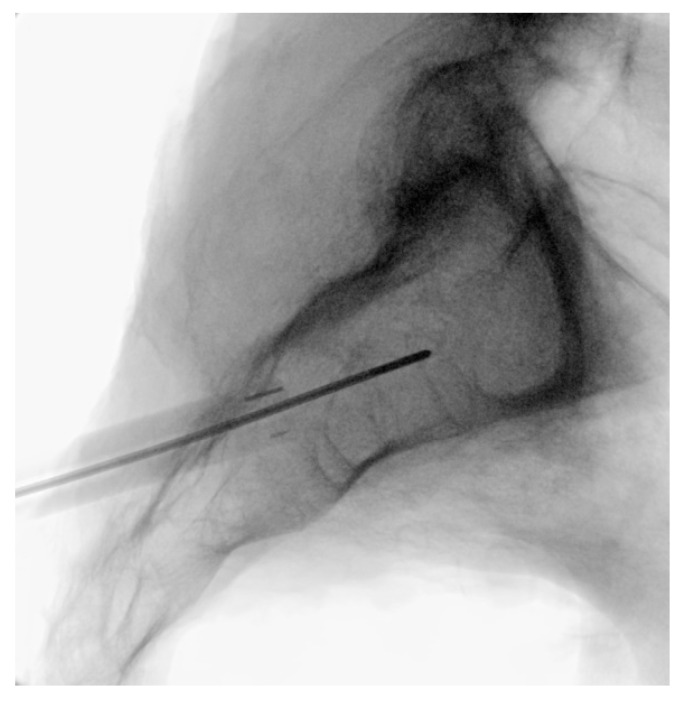
Lateral fluoroscopic view indicating the depth of a properly seated initial dilator.

**Figure 6 jpm-13-01136-f006:**
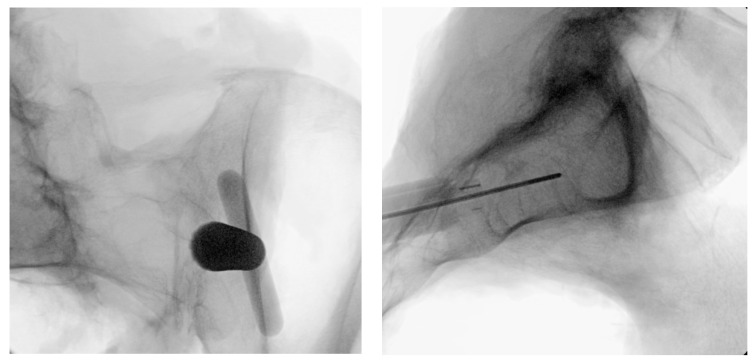
Oblique inlet (**left**) fluoroscopic views are used to verify implant preparation space using a reamer drill visualized in the lateral view (**right**).

**Figure 7 jpm-13-01136-f007:**
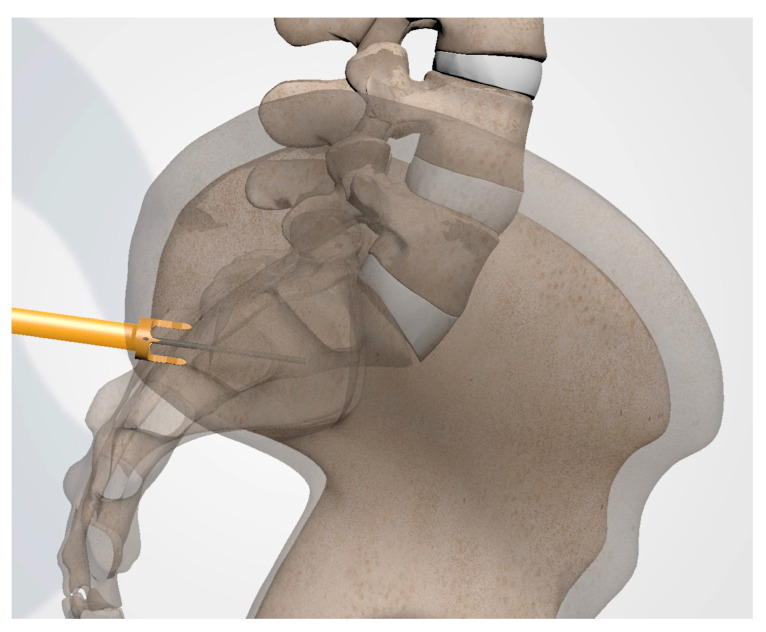
Schematic rendering showing lateral view of docking washer placement.

**Figure 8 jpm-13-01136-f008:**
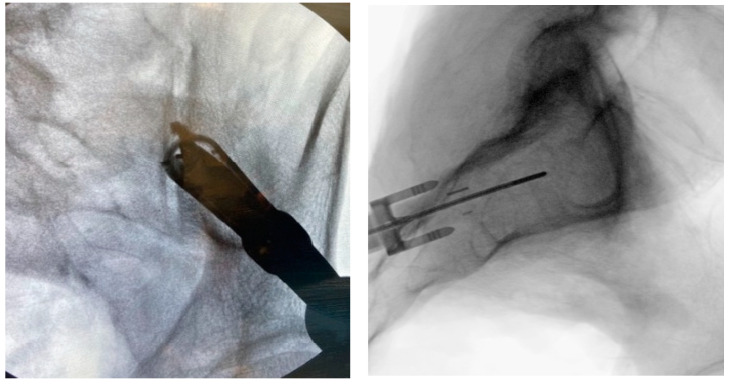
Oblique (**left**), and lateral (**right**) fluoroscopic views are used to confirm docking washer placement, to direct instrumentation and implantation above the sciatic notch.

**Figure 9 jpm-13-01136-f009:**
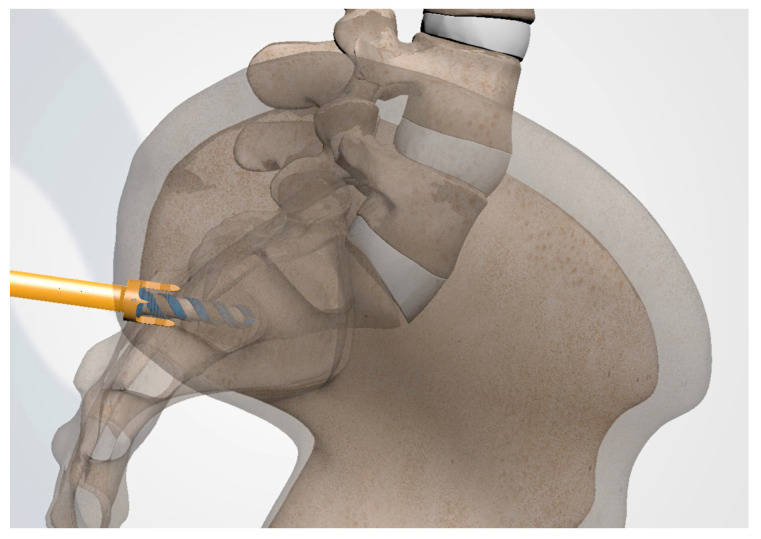
Schematic rendering of proper implant insertion.

**Figure 10 jpm-13-01136-f010:**
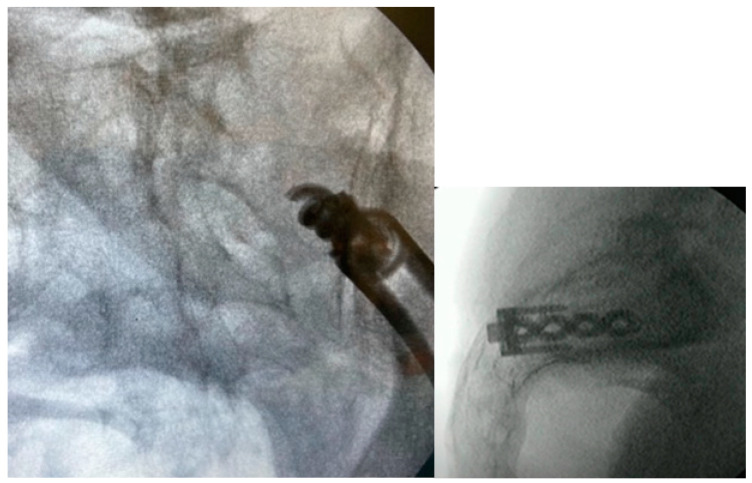
Proper implant insertion and placement showing the implant going through the ilium, across the SI joint, and into the sacrum.

**Figure 11 jpm-13-01136-f011:**
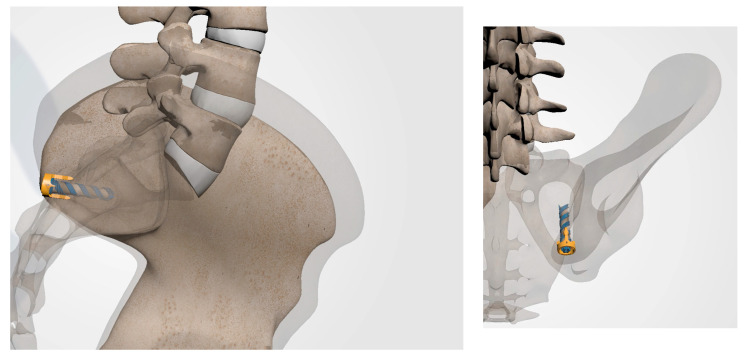
Following removal of all surgical instruments, final imaging is conducted to verify implant placement as illustrated in these anatomical renderings.

**Figure 12 jpm-13-01136-f012:**
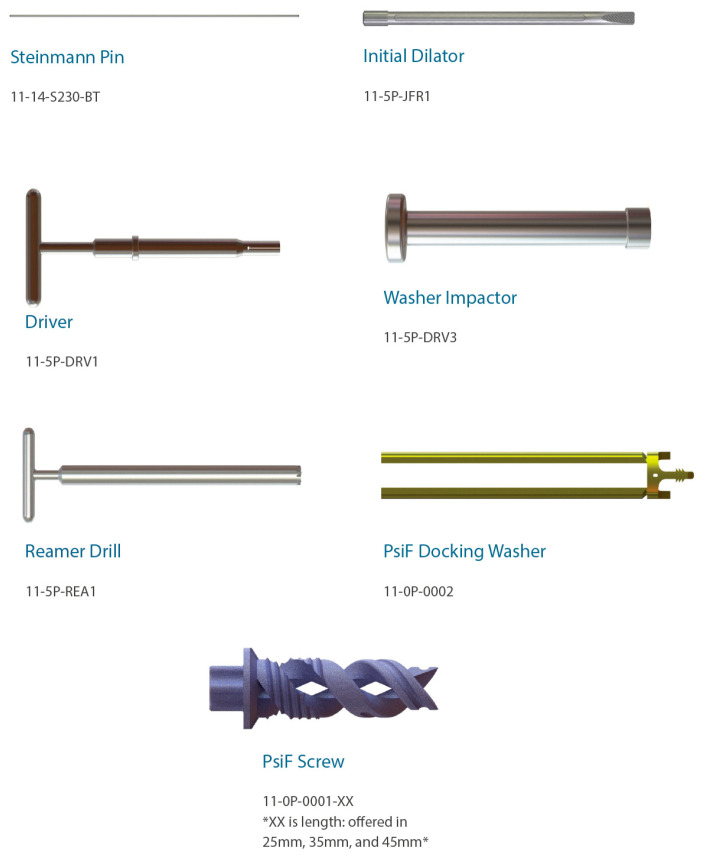
Visual guide of surgical instrumentation.

## Data Availability

Not applicable.

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
