# Peer review of "Minimally Invasive Postero-Inferior Sacroiliac Joint Fusion: Surgical Technique and Procedural Details"

_jpm, 2023, doi:10.3390/jpm13071136_

Round 1

Reviewer 1 Report

This study reports on  a “ ten-step” minimal invasive  surgical procedure associated via a  postero-inferior approach using a  Sacroiliac Joint Fusion System.  The  authors  postulated that implanting the device through the subchondral bone, provides maximum fixation and stabilization of the iliosacral  joint while the  approach to the iliosacral joint from the inferior trajectory also places the implant  perpendicular to the S1 endplate . And  the  authors  concluded  that further l data  from  clinical experience  are encouraged to further validate this procedure as minimally-invasive  method for  SIJ fusion.

There  are  several  reports  published  since  about  10 years  ago  reporting  on  MIS  surgical  techniques.

However Interesting  technical note  on  a  single  MIS  system .  In  the  methods  section I  see  no  details  of  the  surgical technique?   What  anthropometric data  of  the  patients  included  in  this  study?   There  is  no  control  group  for  biomechanical  comparison.  DEXA  was done?

Furthermore, there  are  no  real   actual data  regarding  complications and  fusion rates   since  this  technique  to be  established  clinical  data  and  follow  up  >  2  years  are needed.

Results

“… both the iliac and sacral joint”   I  am  wondering  if  the  authors  could  correct it  to  iliosacral joint?

Figure  1  belongs  to methods  section.

Figure  3  :  “’…the Steinmann  pin has passed the posterior cortical outline of the sacrum, and is docked in the sacrum at or near center of the first sacral body (Figure 3)”    Sorry  but  this Steinman  is  far  below  the  S1  vertebral body.

“the incision will start lateral to the apex of the posterior superior iliac  spine, the incision should also be slightly lateral to the postero-inferior joint line”  I  am  confused  with  this  description/  which  was  the  length  of  the  incision? 

What  is    Under fluoroscopic guidance using the inlet view, advance a Steinmann pin starting inferior,  ventral”  I  cannot  get  it

I  would  recommend  the  authors to  add  a  schematic  representation of  each  step  of  the  surgery.

Figure  7   is   not  sharp, not  clear, for  me.  How   to  get  out  of   trample, to avoid  misguidance ?

Figure   8.   I  see    the   screw-system    the  length  is  rather too short.  Did  the  authors make  a  biomechanical testing  ?

Cost-effectiveness  study?

I  understand  there  was  just  one  patient  in  which  the  system  was  implanted?  Am  I  right ?

English is excellent!

Author Response

Response to Reviewer 1

This study reports on a “ ten-step” minimal invasive  surgical procedure associated via a  postero-inferior approach using a  Sacroiliac Joint Fusion System.  The  authors  postulated that implanting the device through the subchondral bone, provides maximum fixation and stabilization of the iliosacral  joint while the  approach to the iliosacral joint from the inferior trajectory also places the implant  perpendicular to the S1 endplate . And  the  authors  concluded that further l data  from  clinical experience  are encouraged to further validate this procedure as minimally-invasive  method for  SIJ fusion.

There  are  several  reports  published  since  about  10 years  ago  reporting  on  MIS  surgical  techniques.

However interesting  technical note  on  a  single  MIS  system .  In  the  methods  section I  see  no  details  of  the  surgical technique?   What  anthropometric data  of  the  patients  included  in  this  study?   There  is  no  control group  for  biomechanical  comparison.  DEXA  was done?

We appreciate the comment regarding your interest in this technique and procedure.  We elected to include the details of the surgical procedure as separate sections and subsections following the Methods section due to length (see Sections 3.2 to 3.3.9).

This article is solely descriptive of the procedure and contains no patient data.

Furthermore, there  are  no  real   actual data  regarding  complications and  fusion rates   since  this  technique  to be  established  clinical  data  and  follow  up  >  2  years  are needed.

As a procedural description article, this paper does not include long-term patient reported outcomes, complications or imaging data.  Clinical findings with respect to this SIJ fixation system will be forthcoming in subsequent publications.

Results

“… both the iliac and sacral joint”   I  am  wondering  if  the  authors  could  correct it  to  iliosacral joint?

We have made this modification.

Figure  1  belongs  to methods  section.

We have included Figure 1 in subsection 3.1, Structural Anatomy Overview, as it is most germane to this subsection.

Figure  3  :  “’…the Steinmann  pin has passed the posterior cortical outline of the sacrum, and is docked in the sacrum at or near center of the first sacral body (Figure 3)”    Sorry  but  this Steinman  is  far  below  the  S1  vertebral body.

We have altered this text to note the Steinmann pin is near the center of the second sacral body.

“the incision will start lateral to the apex of the posterior superior iliac  spine, the incision should also be slightly lateral to the postero-inferior joint line”  I  am  confused  with  this  description/  which  was  the  length  of  the incision?

The length of the incision is indicated in Section 3.3.5 as 2-3 cm in length.  The current description provides the correct spatial geometry in three dimensions for this approach.

What  is  “  Under fluoroscopic guidance using the inlet view, advance a Steinmann pin starting inferior,  ventral”  I  cannot  get  it

We have re-checked these procedural details, and the current description provides the correct spatial geometry in three dimensions for this approach.

I  would  recommend  the  authors to  add  a  schematic  representation of  each  step  of  the  surgery.

As the procedure is conducted under fluoroscopy, we favor actual real-life fluoroscopic images to renderings of the procedure.

Figure  7   is   not  sharp, not  clear, for  me.  How   to  get  out  of   trample, to avoid  misguidance ?

Figure 7 shows the implant going through the ilium, across the SI joint, and into the sacrum.  This is visualized clearly in the left image, in particular.

Figure   8.   I  see    the   screw-system    the  length  is  rather too short.  Did  the  authors make  a  biomechanical testing  ?

Biomechanical testing data of this implant are being prepared for a subsequent publication.

Cost-effectiveness  study?

We have not conducted a cost-effectiveness study with this implant, although such studies have been conducted for SIJ fusion in general.

I  understand  there  was  just  one  patient  in  which  the  system  was  implanted?  Am  I  right ?

The images provided were drawn from several patients to provide the optimum visualization of each procedural step.

English is excellent!

Thank you.

Reviewer 2 Report

Well prepared and described surgical technique. Introduction can be more clarify. You should attach if possible some pictures from surgery. The discussion should be improved and add some newer references.

Author Response

Response to Reviewer 2

Well prepared and described surgical technique. Introduction can be more clarify. You should attach if possible some pictures from surgery. The discussion should be improved and add some newer references.

We appreciate your comments.  As the procedure is conducted under fluoroscopy, we favored using actual real-life fluoroscopic images of the procedure.  This article is strictly limited to providing the procedural details of this transfixing screw implantation approach.  As such, we have restricted our discussion to the procedure itself; the introductory section provides the requisite current supporting references on SIJ fusion in general.

Reviewer 3 Report

This paper described the surgical procedure associated with the postero-inferior operative trajectory using the PsiF™DNA Sacroiliac Joint Fusion System.

This paper intended to provide the readers with the ten-step surgical procedure regarding The posterior surgical approach with an inferior operative trajectory (postero-inferior) utilizes easily identifiable landmarks to achieve a safest, direct access to the articular joint space for transfixing device placement.

The minimally invasive, transfixing SIJ fusion procedure can employ numerous operative  trajectories. The authors proposed the postero-inferior trajectory allows implant placement in a safe operative corridor, originating at the inferior aspect of the joint, going through the ilium, across the sacroiliac joint space, and into the sacrum, transfixing the osseous confines of the ilium and sacrum across sub-chondral bone.

Even though the authors did not provide a really original way for SI fixation, the ten-step procedure was designed simply and described clearly. The readers could benefit from the method of this communication article.

I recommend this manuscript to be published in your journal.

Author Response

Response to Reviewer 3

This paper described the surgical procedure associated with the postero-inferior operative trajectory using the PsiF™DNA Sacroiliac Joint Fusion System.

This paper intended to provide the readers with the ten-step surgical procedure regarding The posterior surgical approach with an inferior operative trajectory (postero-inferior) utilizes easily identifiable landmarks to achieve a safest, direct access to the articular joint space for transfixing device placement.

The minimally invasive, transfixing SIJ fusion procedure can employ numerous operative  trajectories. The authors proposed the postero-inferior trajectory allows implant placement in a safe operative corridor, originating at the inferior aspect of the joint, going through the ilium, across the sacroiliac joint space, and into the sacrum, transfixing the osseous confines of the ilium and sacrum across sub-chondral bone.

Even though the authors did not provide a really original way for SI fixation, the ten-step procedure was designed simply and described clearly. The readers could benefit from the method of this communication article.

I recommend this manuscript to be published in your journal

We very much appreciate your comments.

Round 2

Reviewer 1 Report

The  authors  have  adressed most of  the  comments. However  my  suggested revision regarding  surgical  tehnique  images  that is  of  major  importance were not  presented,

Author Response

Per the reviewer's request, we have now included several additional schematic renderings (Figures 3,9,10)  that illustrate specific steps in the surgical technique.
